

# Multi-scale hydraulic graph neural networks for flood modelling

Roberto Bentivoglio[1], Elvin Isufi[2], Sebastiaan Nicolas Jonkman[3], and Riccardo Taormina[1]

[1]Department of Water Management, Faculty of Civil Engineering and Geosciences, Delft University of Technology, Delft, The Netherlands
[2]Department of Intelligent Systems, Faculty of Electrical Engineering, Mathematics and Computer Science, Delft University of Technology, Delft, The Netherlands
[3]Department of Hydraulic Engineering, Faculty of Civil Engineering and Geosciences, Delft University of Technology, Delft, The Netherlands

**Correspondence:** Roberto Bentivoglio, r.bentivoglio@tudelft.nl

**Abstract.** Deep learning-based surrogate models represent a powerful alternative to numerical models for speeding up flood mapping while preserving accuracy. In particular, solutions based on hydraulic-based graph neural networks (SWE-GNN) enable transferability to domains not used for training and allow including physical constraints. However, these models are limited due to four main aspects. First, they cannot model rapid differences in flow propagation speeds; secondly, they can

face instabilities during training when using a large number of layers, needed for effective modelling; third, they cannot accommodate time-varying boundary conditions; and fourth, they require initial conditions from a numerical solver. To address these issues, we propose a multi-scale hydraulic-based graph neural network (mSWE-GNN) that models the flood at different resolutions and propagation speeds. We include time-varying boundary conditions via ghost cells, which enforce the solution at the domain's boundary and drop the need of a numerical solver for the initial conditions. To improve generalization over

unseen meshes and reduce the data demand, we use invariance principles and make the inputs independent from coordinates' rotations. Numerical results on dike-breach floods show that the model predicts the full spatio-temporal simulation of the flood over unseen irregular meshes, topographies, and time-varying boundary conditions, with mean absolute errors in time of 0.05 $m$ for water depths and 0.003 $m^2s^{-1}$ for unit discharges. We further corroborate the mSWE-GNN in a realistic case study in The Netherlands and show generalization capabilities with only one fine-tuning sample, with mean absolute errors of 0.12 $m$

for water depth, critical success index for a water depth threshold of 0.05 $m$ of 87.68 %, and speed-ups of over 700 times. Overall, the approach opens up several avenues for probabilistic analyses of realistic configurations and flood scenarios.

## 1 Introduction

Precise flood models are invaluable for evaluating risks, issuing early warnings, and improving preparedness against flood events. Two-dimensional hydrodynamic models determine the spatio-temporal evolution of floods by solving the Shallow

Water Equations (SWE) (Teng et al., 2017). To address the intensive computational demands required to solve the SWE, we can resort to several strategies, such as using simplified physical models (e.g., Van den Bout et al., 2023) and high-performance clusters (e.g., Caviedes-Voullième et al., 2023). More recently, deep learning models emerged as an in-between option that can accelerate flood simulations, while maintaining high accuracy (Bentivoglio et al., 2022). Most deep learning models predict



the flood evolution or its maximum depths while generalizing over different boundary conditions, such as rainfall, on a single

domain. These models include transformers (Pianforini et al., 2024), convolutional neural networks (CNNs) (Berkhahn and Neuweiler, 2024; Liao et al., 2023; Kabir et al., 2020; Guo et al., 2021; He et al., 2023), graph neural networks (GNNs) (Burrichter et al., 2023), Fourier neural operators (Xu et al., 2024), and long short-term memory networks (LSTM) (Wei et al., 2024). Although these methods are effective on a given area, they must be trained again when applied to a different domain, thus hindering their use as surrogate models.

As such, research is now focusing on generalizing deep learning flood models to unseen case studies where the models were not trained on. For example, Löwe et al. (2021), Guo et al. (2022), and Cache et al. (2024) proposed CNN models to estimate the maximum water depth of pluvial floods in urban and catchment settings, respectively. do Lago et al. (2023) and do Lago et al. (2024) developed a conditional generative adversarial network to predict the maximum water depth for unseen rain events and urban catchments. Bentivoglio et al. (2023) proposed a hydraulic-based graph neural network (SWE-GNN)

that could predict the spatio-temporal evolution of dike-breach floods over unseen topographies. The main advantages of this model are its link with finite volume methods that make it suitable to simulate the physics on meshes and a hydraulic-based propagation rule that enforces continuity in water propagation. Moreover, compared to previous works, it can also predict the full flood's spatio-temporal evolution. However, the model cannot reproduce very different propagation speeds and needs a high number of layers when simulating large time steps, which can make the training process unstable. Moreover, this approach

uses a fixed boundary condition and requires the first time step to be given by a numerical solver.

To overcome these limitations, we propose a multi-scale hydraulic graph neural network, based on the SWE-GNN. Multi-scale models combine the domain information coming from different resolutions and have shown benefits for simulating other partial differential equations (Lino et al., 2022; Fortunato et al., 2022). To drop the dependency from the numerical solver, we integrate time-varying boundary conditions via ghost cells, i.e., mesh cells that receive a known value of a given variable

at the domain boundary (LeVeque et al., 2002). To improve the generalization to unseen meshes, we remove all coordinate-dependant inputs. This makes the model invariant to rotations (Bronstein et al., 2021), that is, rotations of the inputs do not affect the outputs. This helps because it prevents the direction of flooding from being biased towards a specific direction in the training data.

We validate the model on dike-breach flood simulations over non-squared domains, discretized by irregular meshes, and

with different topographies and time-varying boundary conditions. To test the applicability of this model to real world case studies, we consider a flood scenario for breaching of a levee system in The Netherlands.

The key novelties of this paper can be summarized as follows:

–  We develop a multi-scale approach which improves the simulations both in speed and accuracy, with speed-ups of up to 1000 times and mean absolute errors of 0.05 $m$ and 0.003 $m^2 s^{-1}$ for water depth and unit discharges, respectively;

–  We include time-varying boundary conditions via the use of ghost cells to remove the dependency from the numerical models and we improve generalization to unseen meshes by making the model's inputs invariant to rotations;





– We show that the model generalizes well to a realistic case study with bigger area and wider range of boundary conditions than the training ones, with only one fine-tuning simulation.

The rest of the paper is organized as follows: in Section 2 we give an overview of the methods, we describe the mesh creation process; introduce the multi-scale hydraulic graph neural network model; describe how to include boundary conditions via ghost cells; detail the inputs and outputs of the model; and present the training loss function. Then, in Section 3 we describe the synthetic and case study datasets and present the results in Section 4. We discuss the method in Section 5 and conclude in Section 6.

## 2 Methodology

We developed a multi-scale graph neural network that combines the information at progressively coarser resolutions to propagate floods in space and time with different flow speeds (Figure 1). The proposed model takes as input static features that represent the topography and connectivity of the domain at different resolutions, and dynamic features that represent the hydraulic variables at time $t$. It then processes them via a U-shaped architecture that applies graph neural networks at different scales and combines them with down-sampling and up-sampling operators. The outputs are the predicted hydraulic variables at following time step $t + 1$ at the finest available resolution. We added boundary conditions by assigning a known value of water depth or discharge to a set of cells at the domain boundary.

In the following, we detail the multi-scale mesh creation procedure (Section 2.1) and the model architecture (Section 2.2). Then, we show how to include boundary conditions (Section 2.3) and rotation-invariant inputs (Section 2.4). Finally, we describe the employed loss function (Section 2.5). We denoted variables $x$ at a given scale or resolution $\mathcal{M}_m$ as $x^{\cdot,m}$, where $\cdot$ is a placeholder for other indices, and where the variable can be a scalar $x$, a vector $\mathbf{x}$, a matrix $\mathbf{X}$, or a tensor $\mathcal{X}$. When the superscript $m$ is omitted, we refer to the variables at the finest scale.

### 2.1 Multi-scale mesh creation

We designed a multi-scale model that combines meshes with progressively coarser resolutions. We employed an iterative process that requires only the boundary polygon of a selected area, without any prior knowledge of the underlying topography. First, we create a coarse mesh from a boundary polygon using the MeshKernel software (Deltares, 2024). This corresponds to the mesh in the bottleneck of the multi-scale module (Figure 1). Then, we refine the mesh by splitting each mesh edge in two and connecting the newly formed points via edges. Then, the mesh undergoes an iterative orthogonalization algorithm needed for the underlying numerical software Delft3d to run because of its staggered grid scheme (Deltares, 2022). If after orthogonalization a mesh edge is too small, then it gets removed, resulting in a mixture of triangular and quadrilateral elements. We repeat these steps multiple times depending on the required scale of computations in the fine mesh. The obtained set of meshes constitutes our multi-scale mesh.

**Multi-scale graph.** The computational graph used in the proposed model considers as nodes the barycenters of the mesh cells, while edges connect neighbouring cells. We connect the graphs at two scales based on the spatial position of the mesh



**Figure 1.** Overview of the proposed mSWE-GNN model. The model $\Phi(\cdot)$ takes as input a fine mesh and its coarser versions, along with the static and dynamic inputs defined on them (blue box, top left) and produces an estimate of the hydraulic variables in time (orange box, top right). The model is then repeated auto-regressively using its predictions as inputs (top black arrow), to determine the spatio-temporal evolution of the flood. Boundary conditions are provided at each time step by assigning a known value to a set of cells in the dynamic inputs $\mathbf{U}^{t-p:t}$. In the black box, black arrows indicate multi-layer perceptrons (MLP) present in the encoders and the decoder; blue arrows represent graph neural network layers; light green arrows down-sampling layers; dark green arrows up-sampling layers; and red arrows skip connections across different parts of the architecture.



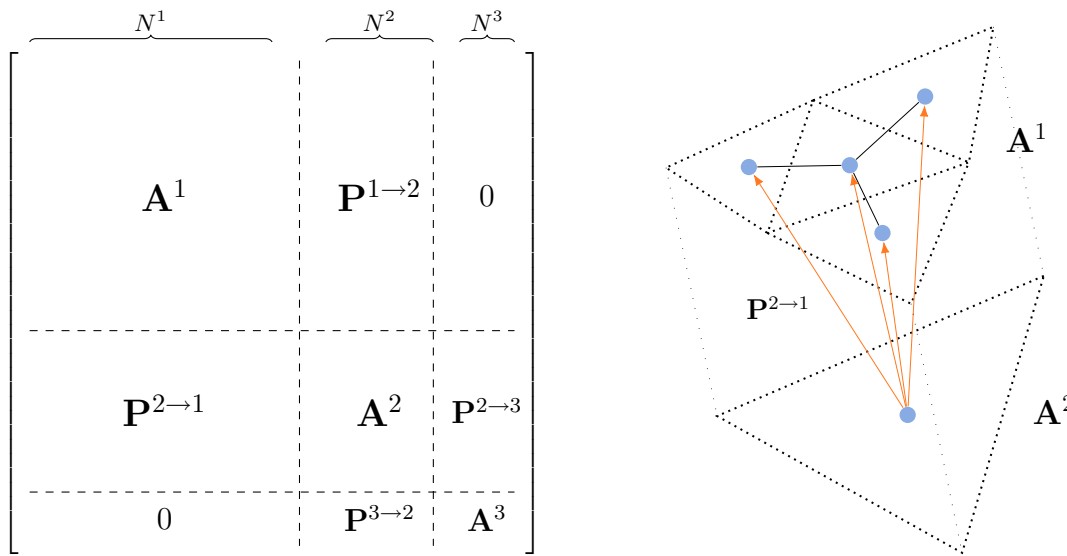

**Figure 2.** Left: adjacency matrix representation of the multi-scale graph, for a mesh with three scales. $\mathbf{A}^m \in \mathbb{R}^{N^m \times N^m}$ represents the adjacency matrix at scale $m$, while $\mathbf{P}^{m \to n} \in \mathbb{R}^{N^m \times N^n}$ represent the prolongation matrix from scale $m$ to scale $n$. Right: example connection between a fine mesh $\mathbf{A}^1$ and a coarse mesh $\mathbf{A}^2$, where $\mathbf{P}^{2 \to 1}$ indicates the connectivity across the two scales.

barycenters, as shown in Figure 2. If a fine mesh cell's center is within a coarse mesh cell center, then a directed inter-scale

90 edge exists between the two nodes.

We can describe the connectivity of the obtained multi-scale graph via a block-diagonal connectivity matrix composed by adjacency matrices $\mathbf{A}^m$ and prolongation matrices $\mathbf{P}^{m \to n}$. Adjacency matrices are squared matrices that represent the connectivity of a graph at scale $m$ by assigning $a_{ij}^m = 1$ if edge $(i, j) \in \mathcal{E}^m$. Prolongation matrices are rectangular matrices that act like adjacency matrices, but connect one scale $m$ to its upper or lower scale $n \in \{m+1, m-1\}$. They can also be seen as

95 adjacency matrices for bipartite graphs whose nodes can be divided into two disjoint sets.

## 2.2 Architecture

We develop the Multi-scale Hydraulic Graph Neural Network (mSWE-GNN) by building upon Bentivoglio et al. (2023). This is an encoder-processor-decoder architecture that auto-regressively predicts the hydraulic variables at time $t+1$ as

$$\hat{\mathbf{U}}^{t+1} = \mathbf{U}^t + \Phi(\mathbf{X}_s, \mathbf{U}^{t-p:t}, \mathcal{E}), \tag{1}$$

100 where the output $\hat{\mathbf{U}}^{t+1}$ corresponds to the predicted hydraulic variables, $\mathbf{U}^t$ are the hydraulic variables (water depth $[m]$ and unit discharge $[m^2 s^{-1}]$) at time $t$, $\Phi(\cdot)$ is an encoder-processor-decoder model that determines the evolution of the hydraulic variables for a fixed time step, $\mathbf{X}_s$ are the static node features, $\mathbf{U}^{t-p:t}$ are the dynamic node features, i.e., the hydraulic variables for time steps $t - p$ to $t$, and $\mathcal{E}$ are the edge features that describe the geometry of the mesh. We include different mesh





resolutions by defining the model $\Phi(\cdot)$ in a U-shaped architecture, inspired by Gao and Ji (2019), starting from a fine mesh to
coarser ones and back to a fine mesh output. Hereafter, we describe the details of the architecture shown in Figure 1.

**Encoder.** We increase the expressivity of the inputs by employing three separate encoders for processing the static node
features $\mathbf{X}_s \in \mathbb{R}^{N \times I_s}$, the dynamic node features $\mathbf{X}_d \equiv \mathbf{U}^{t-p:t} \in \mathbb{R}^{N^1 \times O(p+1)}$, and the edge features $\boldsymbol{\mathcal{E}} \in \mathbb{R}^{E \times I_\varepsilon}$, with $N$ the
total number of nodes, $I_s$ the number of static node features, $N^1$ the number of nodes at the finest scale, $O$ the number of
hydraulic variables, $p$ the number of input previous time steps, $E$ the number of edges, and $I_\varepsilon$ the number of input edge
features. The encoded variables are defined as

$$\mathbf{H}_s = \phi_s\left(\mathbf{X}_s\right), \mathbf{H}_d = \phi_d\left(\mathbf{X}_d\right), \boldsymbol{\mathcal{E}}' = \phi_\varepsilon\left(\boldsymbol{\mathcal{E}}\right), \tag{2}$$

where $\phi_s(\cdot)$ and $\phi_d(\cdot)$ are 3-layer MLPs shared across all nodes and $\mathbf{H} \in \mathbb{R}^{N \times G}$ the encoded node features; $\phi_\varepsilon(\cdot)$ is a 3-layer
MLP shared across all edges that encodes the edge features into $\boldsymbol{\mathcal{E}}' \in \mathbb{R}^{E \times G}$, and $G$ the number of features in the latent space.
We apply the shared encoders of the static features $\phi_s(\cdot)$ and $\phi_\varepsilon(\cdot)$ to all features at all scales, while the encoder of the dynamic
features $\phi_d(\cdot)$ is applied only to the finest scale. The rationale behind having a shared static feature encoder for all scales is that
higher-dimensional features should have a similar embedding independently of the scale, since the physical quantities are the
same.

**Processor.** The processor propagates the encoded inputs throughout the multi-scale graph. We employ a sequence of GNN
layers to propagate information at a given scale and connect two scales via down-sampling and up-sampling operators. The
operations are organized in a U-shaped fashion, with a down-sampling branch from fine to coarse and a up-sampling branch
from coarse to fine, as shown in Figure 1.

In the down-sampling branch, we start by applying $L$ GNN layers at the encoded node and edge features $\mathbf{H}_s^1$, $\mathbf{H}_d^1$, and $\boldsymbol{\mathcal{E}}'^1$
at the finest-scale mesh $\mathcal{M}_1$. Then, we apply a down-sampling operator $\downarrow\colon \mathcal{M}_m \to \mathcal{M}_{m+1}$ that maps the features of the finer
scale $\mathcal{M}_m$ to the coarser scale $\mathcal{M}_{m+1}$. We repeat these two operations until the final coarser scale. In the up-sampling branch,
we apply an up-sampling operator $\uparrow\colon \mathcal{M}_{m+1} \to \mathcal{M}_m$ that maps the features from the coarser scale $\mathcal{M}_{m+1}$ to the finer scale
$\mathcal{M}_m$. We add skip connections to sum the output of the down-sampling GNN at scale $\mathcal{M}_m$ with the output of the up-sampling
operator from scale $\mathcal{M}_{m+1}$ to $\mathcal{M}_m$. These connections facilitate information transfer and training, similarly to Ronneberger
et al. (2015). Finally, we apply another set of $L$ GNN layers to the output of the skip connections and repeat these operations
until the finest scale. All GNNs, down-sampling operators, and up-sampling operators are not shared, meaning that each acts
independently at one scale or across two given scales.

The GNN layers follow Bentivoglio et al. (2023) and can be expressed as

$$\mathbf{s}_{ij}^{(\ell+1)} = \psi\left(\mathbf{h}_{si}, \mathbf{h}_{sj}, \mathbf{h}_{di}^{(\ell)}, \mathbf{h}_{dj}^{(\ell)}, \boldsymbol{\varepsilon}_{ij}'\right) \odot \left(\mathbf{h}_{dj}^{(\ell)} - \mathbf{h}_{di}^{(\ell)}\right), \tag{3}$$

$$\mathbf{h}_{di}^{(\ell+1)} = \mathbf{h}_{di}^{(\ell)} + \sum_{j \in \mathcal{N}_i} \mathbf{s}_{ij}^{(\ell+1)} \mathbf{W}^{(\ell+1)}, \tag{4}$$

where $\psi(\cdot) \colon \mathbb{R}^{5G} \to \mathbb{R}^G$ is an MLP, $\odot$ is the Hadamard (element-wise) product, and $\mathbf{W}^{(\ell)} \in \mathbb{R}^{G \times G}$ are learnable parameter
matrices. The propagation rule in Eq. (3) has a hydraulic gradient-like term, $\mathbf{h}_{dj}^{(\ell)} - \mathbf{h}_{di}^{(\ell)}$, that acts as a physical constraint that



allows water to propagate only from nodes which already have water. In fact, $\mathbf{h}_{di} = 0$ only if node $i$ has both zero water depth and discharge, since the dynamic node encoder has no bias term. The predicted fluxes across nodes $\mathbf{s}_{ij}$ then combine the information from neighbouring nodes $\mathcal{N}_i$ by following the principles of numerical methods.

The down-sampling operator $\downarrow : \mathcal{M}_m \to \mathcal{M}_{m+1}$ is a mean pooling operator[1] from a fine mesh $\mathcal{M}_m$ to a coarse mesh $\mathcal{M}_{m+1}$ defined as

$$\mathbf{h}_{di}^{m+1} \leftarrow \frac{1}{|\mathcal{N}_i^{m \to m+1}|} \sum_{\mathcal{N}_i^{m \to m+1}} \mathbf{h}_{di}^m, \tag{5}$$

where $\mathcal{N}_i^{m \to m+1}$ is the set of neighbouring nodes in the finer mesh $\mathcal{M}_m$ connected vertically to the nodes in the coarser mesh $\mathcal{M}_{m+1}$ and $\mathbf{h}_{di}^{m+1} \in \mathbb{R}^G$ are the down-sampled dynamical features at node $i$. We used a mean pooling operation since physical features at coarser scales should resemble those at the finer scale. This approach offers a trade-off between simpler resampling methods such as nearest neighbour and more computationally intensive ones such as cubic interpolation (Maeland, 1988).

The up-sampling operator $\uparrow : \mathcal{M}_{m+1} \to \mathcal{M}_m$ is a learnable operator defined as

$$\mathbf{h}_{di}^m \leftarrow \sum_{\mathcal{N}_i^{m+1 \to m}} \psi^{m+1 \to m} \left( \mathbf{h}_{si}^m, \mathbf{h}_{si}^{m+1}, \mathbf{h}_{di}^m, \mathbf{h}_{di}^{m+1} \right) \odot \mathbf{h}_{di}^{m+1}, \tag{6}$$

where $\mathbf{h}_{di}^m$ are the up-sampled dynamic node features at node $i$ in scale $\mathcal{M}_m$, $\psi^{m+1 \to m}(\cdot) : \mathbb{R}^{4G} \to \mathbb{R}^G$ is an MLP, and $\mathcal{N}_i^{m+1 \to m}$ is the set of neighbouring nodes in the coarser mesh $\mathcal{M}_{m+1}$ to the nodes in the finer mesh $\mathcal{M}_m$. This expression has two important features: first, it is independent of the number of nodes in the fine scale, meaning that it works both from one-to-one node or from one-to-several nodes; second, the multiplication by $\mathbf{h}_{di}^{m+1}$ ensures that this operation only activates when a node on the coarse cell has water in it, i.e., $\mathbf{h}_{di}^{m+1} \neq 0$. Differently from the SWE-GNN layer (Eq. (3)), we avoid edge features, since there are none across scales, and the hydraulic gradient term since the values at one scale should be close to those at the previous scale. Thus, using a difference would result in a zero value when the features at two scales are identical.

We add skip connections to combine the outputs of the down-sampling GNNs $\mathbf{h}_{di}^{m\downarrow}$ with the outputs of the up-sampling operations $\mathbf{h}_{di}^{m\uparrow}$, before applying another GNN layer. The skip connections can be expressed as

$$\mathbf{h}_{di}^m \leftarrow \mathbf{h}_{di}^{m\downarrow} + \mathbf{h}_{di}^{m\uparrow}. \tag{7}$$

Skip connections should improve the connectivity between different parts of the architecture and combine the different propagation speeds.

The obtained mSWE-GNN architecture allows us to model the flood's propagation speed at a different scales. This is because each scale's GNN covers different portions of space based on physical nodes' distances. These separate flow speeds are combined in the architecture allowing the model to capture better their variations from one time step to another. This is particularly relevant for capturing a broader scale of dynamics with rapidly time-varying boundary conditions that change significantly the propagation speed. Moreover, this setup alleviates the requirements on the number of GNN layers at the finest scale since one

---

[1]We also evaluated a learnable pooling operator, but the performance was lower, as highlighted in the ablation study in Sec. 4.3.



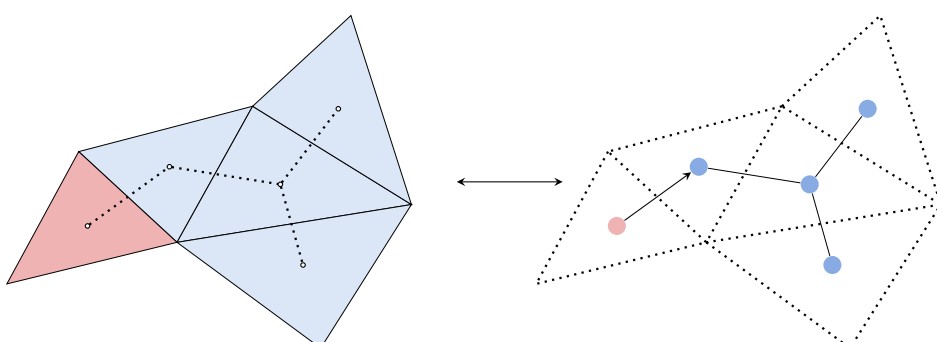

**Figure 3.** Schematic representation of an arbitrary triangular volume mesh with a ghost cell (left). The ghost cell (red) is added in correspondence of a boundary cell which receives a given boundary condition. In the dual graph (right), a directed edge is added from the ghost cell to the domain cell, or vice-versa, depending on the type of boundary condition.

layer at a coarse scale can cover the equivalent of several layers at the finest scale. Hence, we end up with a model that is more efficient and better captures the time-varying dependencies of the flood.

**Decoder.** The decoder estimates the predicted hydraulic variables $\hat{\mathbf{u}}_i^{t+1} \in \mathbb{R}^O$ as a combination of the input previous time steps $\mathbf{U}_i^{t-p:t} \in \mathbb{R}^{O \times (p+1)}$ and the output of the processor at the finest scale $\mathbf{h}_{di} \in \mathbb{R}^G$. This can be expressed as

$$\hat{\mathbf{u}}_i^{t+1} = \text{ReLU}\left(\mathbf{U}_i^{t-p:t}\mathbf{w}_p + \varphi\left(\mathbf{h}_{di}\right)\right), \tag{8}$$

where $p$ is the number of previous time steps, $\mathbf{w}_p \in \mathbb{R}^{p+1}$ is a learnable vector, and $\varphi(\cdot)$ is a 3-layer MLP which decodes the embeddings of the processor $\mathbf{h}_{di}$. We added a ReLU $=\max\{0, x\}$ activation function at the output of the decoder to guarantee physical values of water depths, since we know that water depth and unit discharges cannot be negative, similarly to Palmitessa et al. (2022). The learnable parameters $\mathbf{w}_p$ weigh the contribution of each input time step to the output of the model, thus acting as a 1D convolutional layer along the temporal axis.

## 2.3 Boundary conditions

To include external forcings, we add boundary conditions via ghost cells, as done in numerical methods (LeVeque et al., 2002). Ghost cells are elements which belong to the computational domain but not in the physical one and act as link to external conditions. Boundary conditions related to inflows and outflows are represented via directed edges towards the real mesh and the ghost cells, respectively, as shown in Figure 3. The computations with directed edges in the model follow the same propagation rules as for undirected edges. Based on the forcing type, we can assign a prescribed condition for each time step of the simulation and at a specific point in the domain, to strictly enforce boundary conditions. For water levels, we impose the known value at the boundary. For discharge hydrographs, we first transform discharges $[m^3 s^{-1}]$ into unit discharges $[m^2 s^{-1}]$, by dividing the input discharge by the length of the edge across which it is passing, as in numerical methods. Wall boundaries are modelled without any ghost cell instead of imposing reflection since this is implicitly assumed by the dual graph's structure that cannot propagate over the wall.





## 2.4    Rotation-invariant inputs

Most deep learning models consider coordinate-dependent features, such as the $x$ and $y$ components of the slopes. When applying a rotation to a domain, these values change, causing a change in the output, which is not necessarily equivalent to the applied rotation. This is a well-studied challenge in DL models (Bronstein et al., 2021) and can be solved via data augmentation,

i.e., by training the model using rotated instances of the training simulations, or by modifying the deep learning model (e.g., Lino et al., 2022). Since the outputs of our model are scalars, we avoid using any rotation-dependant features to simplify the model and obtain a rotation-invariant model, i.e., rotations of the inputs do not affect the output. The static node features can then be expressed as $\mathbf{x}_{si} = (a_i, e_i, m_i, w_i^t)$, where $a_i$ is the area of the $i^{th}$ finite volume cell, $e_i$ its elevation, $m_i$ its Manning coefficient, and $w_i^t$ its water level. To determine the values of elevation $e_{i,m}$, Manning coefficient $m_{i,m}$, and water level $w_{i,m}^t$

at the coarser scales, we perform a mean pooling operation from the finest scale to each of the coarser scales as in Eq. (5). As edge features, we consider $\boldsymbol{\varepsilon}_{ij} = (l_{ij})$, where $l_{ij}$ is the length of the dual edge between node $i$ and node $j$. The dynamic node features are defined as $\mathbf{x}_{di} = \mathbf{u}_i^{t-p:t} = (\mathbf{u}_i^{t-p}, ..., \mathbf{u}_i^{t-1}, \mathbf{u}_i^t)$, with $\mathbf{u}_i^t = (h_i^t, |q|_i^t)$, where $h_i^t$ is the water depth at time $t$ and node $i$, and $|q|_i^t$ is the unit discharge at time $t$ and node $i$.

## 2.5    Loss function

We employ a multi-step-ahead forecasting loss $\mathcal{L}_f$ that considers multiple model's outputs using its own predictions as inputs. This helps the model dealing with incorrect inputs and is useful to reduce accumulation of errors in time (Bentivoglio et al., 2023). It can be expressed as

$$\mathcal{L}_f = \frac{1}{HO} \sum_{\tau=1}^{H} \sum_{o=1}^{O} \gamma_o \|\hat{\mathbf{u}}_o^{t+\tau} - \mathbf{u}_o^{t+\tau}\|_2, \tag{9}$$

where $\mathbf{u}_o^{t+\tau} \in \mathbb{R}^N$ are the predicted hydraulic variables at time $t+\tau$, $H$ is the prediction horizon, and $\gamma_o$ are coefficients used

to weigh the influence of each hydraulic variable to the loss.

## 3    Experimental setup

### 3.1    Synthetic dataset

We created a synthetic dataset of dike-breach flood simulations, using the numerical software Delft3d (Deltares, 2022). Each simulation is discretized via an irregular mesh created from randomly generated polygons, based on ellipsoidal shapes, as

described in Sec. 2.1. The multi-scale mesh obtained with this procedure has a total of four scales. This is an arbitrary choice selected to showcase the expressivity of the model, but different number of mesh scales would work as well, unless the coarsest scale has excessively few cells. For each mesh, we use a randomly generated digital elevation model (DEM), based on Perlin noise and combined with a small slope in a random direction, as exemplified in Figure 4. As boundary condition, we apply an inflow discharge hydrograph on one random border edge. The hydrograph's shape is generated based on Weibull-like



**Table 1.** Mean and standard deviation of elevation (above sea level), number of cells, cell area, edge length, and total flood volume for the training, validation, and testing datasets.

| Dataset | # Simulations | Elevation [$m$] | Number of cells | Cell area [$m^2$] | Edge length [$m$] | Flood volume [$10^6 m^3$] |
|---------|---------------|-----------------|------------------|--------------------|--------------------|-----------------------------|
| Train | 60 | -0.04 ± 0.6 | 10018 ± 1251 | 14817 ± 5717 | 182.8 ± 37.2 | 3.07 ± 0.66 |
| Validation | 20 | -0.06 ± 0.58 | 10029 ± 904 | 13741 ± 5125 | 176.3 ± 34.9 | 2.9 ± 0.69 |
| Test | 20 | -0.03 ± 0.53 | 9803 ± 1130 | 13480 ± 4917 | 174.9 ± 33.7 | 3.02 ± 0.64 |
| Test dike ring 15 | 10 | -1.07 ± 1.17 | 22881 | 13544 ± 5521 | 174.7 ± 36.9 | 26.5 ± 2.54 |

probability density functions with different shape parameters (Bhunya et al., 2011). All hydrographs are right-tailed since most dike-breach hydrographs have this shape (e.g. D'Oria et al., 2022; Shustikova et al., 2020) and their peaks vary from 150 to 300 $m^3 s^{-1}$, as shown in Figure 6, in line with realistic breach inflows. The dataset comprises 100 simulations, 60 used for training, 20 for validation, and 20 for testing. Each simulation has as output a temporal resolution of two hours for a total simulation time of 96 hours, or 48 steps ahead. The datasets' statistics in terms of elevation (above sea level), number of cells, cell area,

edge length, and total flood volume are reported in Table 1. Compared to the dataset in Bentivoglio et al. (2023), this has more complexity, both in terms of mesh structure and discharge conditions.

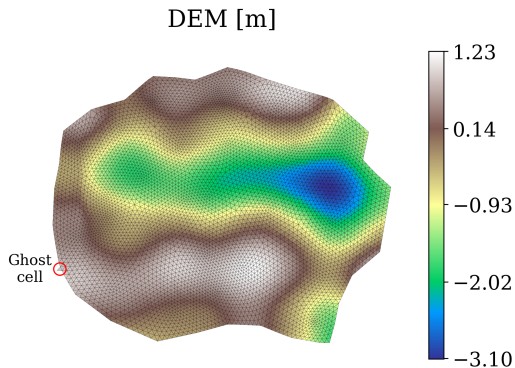

**Figure 4.** Example mesh with the corresponding digital elevation model (DEM) for one simulation in the synthetic dataset.

## 3.2 Case study: dike ring 15

We assess the transferability of the trained model by applying it to dike ring 15 Lopiker en Krimpenerwaard in The Netherlands, which surrounds and protects the area between Rotterdam and Utrecht (Figure 5). This area is prone to flooding and protected

entirely by a system of levees. This case study has an area of 31400 $ha$, with a total population of 201,500 inhabitants and an expected flood damage per event of 5.1 billion euros (Boon and Witteveen+Bos, 2011). We chose this area because, depending on the location of the breach, the basin has a bathtub or sloped response, meaning that water fills up the domain evenly or has a



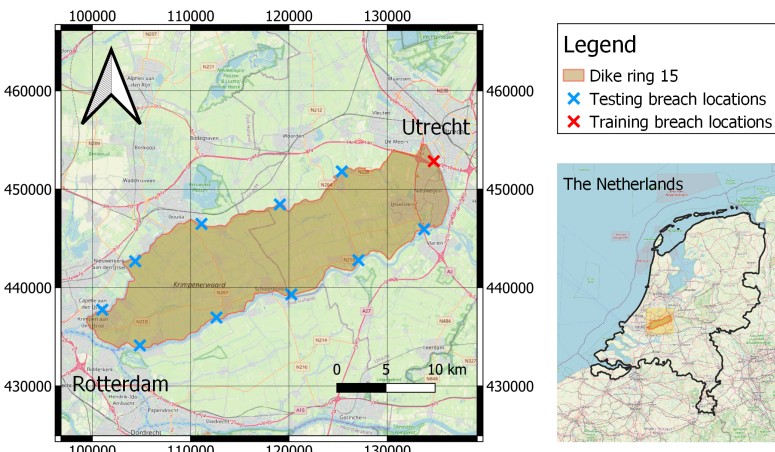

**Figure 5.** Dike ring 15, in The Netherlands (coordinate system EPSG:28992 - Amersfoort / RD New). The crosses indicate the location of the dike breaches used for training and testing. The maps are taken from ©OpenStreetMap contributors 2024. Distributed under the Open Data Commons Open Database License (ODbL) v1.0.

preferential drainage direction, respectively (Rijkswaterstaat, 2014). We simplified the hydraulic components not represented in the training dataset. Specifically, we removed all water bodies and every infrastructure that is not directly included in the

DEM. Moreover, we assumed constant roughness coefficients throughout the whole area.

As boundary conditions, we created a set of inflow discharges that follow a different distribution from the training ones. This has an initial rise, following an hypothetical widening of the breach, and a decreasing limb in time that ends with non-zero discharge. We also increased the peak discharge to match realistic values for the considered case study, with values between 700 and 1000 $m^3 s^{-1}$, corresponding to inflows of a fully developed breach, which could be more than 100m wide. This results

in an increase of the total flood volumes by approximately nine times with respect to the synthetic dataset. For the breaching locations, we selected 11 approximately equidistant spots along the contour of the dike ring (see Figure 5). This allows us to capture a comprehensive hydraulic responses of the basin.

The selected case study is more than twice as big as the synthetic datasets and has different elevation patterns, leaving more space to develop different flood dynamics. Hence, we decided to test the model also with a fine-tuning step, employing a single

simulation for training and validation. In the experiments, we analyse the effect of adding this fine-tuning step after training the model on the synthetic dataset.

### 3.3   Normalization

The static attributes (node and edge features) are determined at all scales when creating the dataset. Since the values of areas $a$ and edge lengths $l$ change significantly across scales, we standardize those features separately for each scale. Specifically,

we collect all training instances of a given variable $x$ at mesh scale $\mathcal{M}_m$ and determine their mean $\mu$ and variance $\sigma$. The





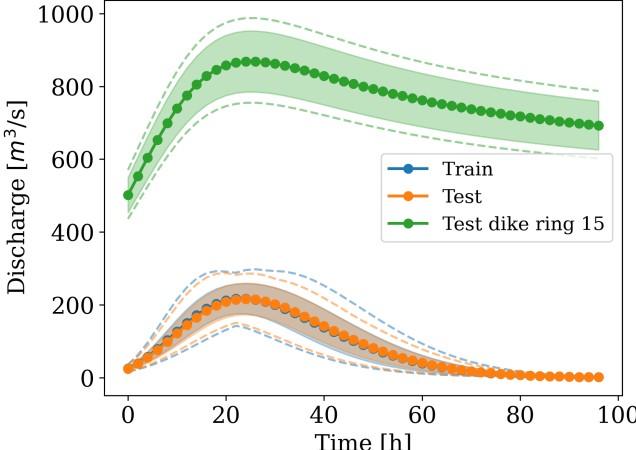

**Figure 6.** Distribution and shape of the hydrographs used as inputs for the training (blue), synthetic test (orange), and dike ring 15 test (green) simulations. The shaded region indicates one standard deviation away from the mean, at each time step. The dotted lines represent the envelopes of the minimum and maximum discharges at each time step.

normalized variables are then obtained as $\hat{x} = \frac{x-\mu}{\sigma}$, where $\hat{x}$ indicates the standardized variable. The remaining variables are not processed by any normalization procedure.

### 3.4 Training setup

We trained all models with Pytorch (Version 2.0.1) (Paszke et al., 2019) and Pytorch Geometric (Version 2.4) (Fey and Lenssen, 2019) libraries, using the Adam optimization algorithm (Kingma and Ba, 2014). We performed several preliminary trials to identify a set of suitable training hyperparameters for the experiments; see Table D1. We used a learning rate scheduler with a fixed step decay of 0.7, every 20 epochs, starting from 0.003. The training was carried out for 200 epochs with early stopping, using 16-bit mixed-precision to decrease the computational burden. During training, we clipped the gradients with a value higher than 1, to improve training stability, and employed a curriculum learning strategy as in Bentivoglio et al. (2023), with a maximum training prediction horizon $H = 6$ steps ahead (Eq. (9)). We used $p = 2$ previous time steps as dynamic inputs, i.e., $\mathbf{X}_d = (\mathbf{U}^{t-2}, \mathbf{U}^{t-1}, \mathbf{U}^t)$. The coefficients used in the loss function (Eq. (9)) are $\gamma_1 = 1$ for the weight of the water depths, and $\gamma_2 = 7$ for the weight of the unit discharge. We used these values to weight more water depths, which values are generally more than 10 times larger than the discharge ones, as we deem them more important.

In terms of hardware, we employed an NVIDIA A100 80GB PCIe (Delft High Performance Computing Centre , DHPC) for training and deployment of the deep learning models, and an Intel(R) Core(TM) i7-8665U @1.9 GHz CPU for the execution of the numerical model. Note that the numerical model cannot run on GPUs, but we used the available OpenMP option to parallelize the computations on 8 CPU threads.



## 3.5 Metrics

We evaluated the models' performance using a multi-step-ahead mean absolute error (MAE) for each hydraulic variable $\hat{\mathbf{u}}_o^\tau$

over the full simulation, expressed as:

$$MAE_o = \frac{1}{H} \sum_{\tau=1}^{H} \|\hat{\mathbf{u}}_o^\tau - \mathbf{u}_o^\tau\|_1, \tag{10}$$

with $H$ being the prediction horizon. Note that while the training loss in Eq. (9), is evaluated over a limited number of time steps, the validation loss function in Eq. (10) is evaluated on the full simulation, to mimic the testing conditions.

We also measured the spatio-temporal error distribution of the water depth using the critical success index (CSI) for threshold

values of 0.05 m and 0.3 m as in Bentivoglio et al. (2023). The CSI measures the spatial accuracy of detecting a certain class (e.g., flood or no-flood) and for a given threshold it is evaluated as

$$CSI = \frac{TP}{TP + FP + FN} \tag{11}$$

where TP are the true positives, i.e., number of cells where both model and simulations predict flood, FP are the false positives, i.e., number of cells where the model wrongly predicts flood, and FN are the false negatives, i.e., number of cells where the

model does not recognize a flooded area. We measured the computational speed-up as the ratio between the computational time required by the numerical model and the inference time of the deep learning model. We did not consider the computational time to create the meshes, since they are needed for both methods. Unless otherwise mentioned, the deep learning model is run in parallel over all testing simulations, differently from the numerical model (see Appendix C). This choice is reasonable since we can use this model for probabilistic forecasts, where multiple simulations may be run in parallel.

# 4 Results

## 4.1 Comparison with SWE-GNN

To highlight the improvements given by multi-scale modelling, we compared the mSWE-GNN model with an enhanced SWE-GNN model that includes ghost cells, rotation-invariant inputs, and the 1D CNN in the decoder, but lacks the multi-scale component. We did not compare with the standard SWE-GNN since it would not be able to run without a numerical input.

We also did not compare against other baselines as the SWE-GNN performs better than them (Bentivoglio et al., 2023). Both models underwent a hyperparameter search procedure based on the number of GNN layers and the number of hidden features, as reported in Table D1.

This resulted in a set of models with different performances in terms of accuracy and speed as reported in Figure 7. The results show that the multi-scale structure helps the model to better capture flow variations across time, resulting in a better

Pareto front for validation losses and CSI. The mSWE-GNN has on average more parameters than the SWE-GNN because it has several GNNs (two per each scale, except one for the bottleneck) which makes it by default bigger. Despite this, the mSWE-GNN is comparatively faster, with speed-ups of up to 1200 times, since at the finest scale it has fewer layers than the



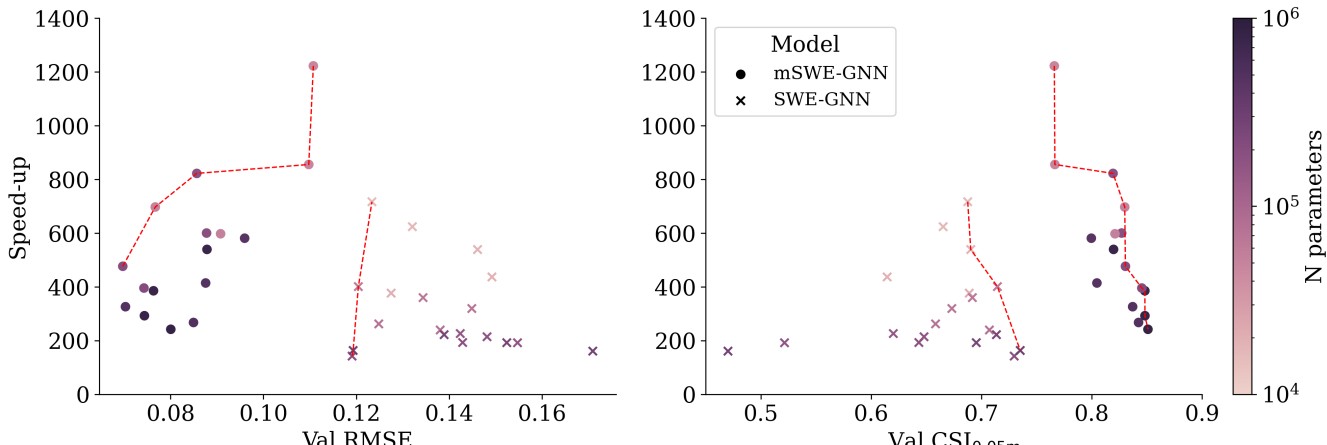

**Figure 7.** Pareto front of the mSWE-GNN and SWE-GNN models for speed-ups, validation RMSE (left), and validation CSI with $0.05m$ water depth threshold (right). The models' size varies with number of hidden features and number of GNN layers.

SWE-GNN. This reduces substantially the computations since the finest scale is the one with most nodes and edges. Moreover, the training process resulted also more stable in the mSWE-GNN, probably due to the lower number of GNN layers.

For the remaining analyses, we selected the mSWE-GNN model with the best performance, which consists of 4 GNN layers for each scale, a hidden feature dimension of 64, and around 811k learnable parameters. Despite the limited amount of training samples and the amount of variability in simulated conditions, the model captures the flow patterns. Figure 8 reports the evolution of the critical success index (CSI) for the water depth thresholds of $0.05m$ and $0.3m$ and the mean absolute errors (MAE) for water depth and unit discharge for the test dataset. The $\text{CSI}_{0.05m}$ stays constantly high for all simulations.

On the other hand, the $\text{CSI}_{0.3m}$ starts low: this is due to an initial scarcity of water depths higher than $0.3m$, which skews the performance to lower values. The MAE of unit discharge seems correlated with the input breach discharge values, meaning that the biggest errors are in correspondence of the hydrograph peak and the smallest in correspondence of the tail. Indeed, the highest errors are generally located in correspondence of the breach location, where the most rapid processes occur. Thus, when the inflow discharge decreases, so does the error. The MAE of water depth instead rises with time as also reported in

Bentivoglio et al. (2023). In this case, however, the errors plateau in correspondence of the end of the inflow hydrograph, indicating that water flow is stopping.

## 4.2   Transfer learning to realistic case study

After training the model with the synthetic dataset, we tested it on dike ring 15, for different breach locations with varying discharges. The zero-shot testing of the model without any fine tuning resulted in modest performances in Table 2. We attribute

this mismatch to the difference in total flood volume but also the domain size and the different elevation patterns when compared to the training ones (see Table 1). Moreover, this implies different hydraulic dynamics, such as the presence of sloped



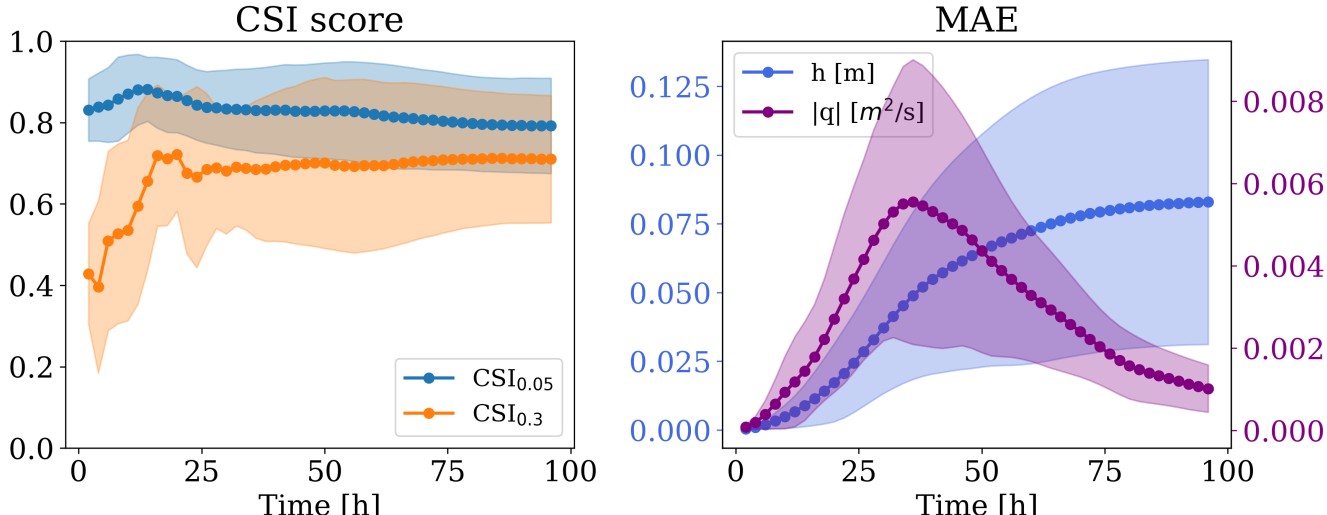

**Figure 8.** Temporal evolution of CSI scores (left) and MAE of water depth $h$ and unit discharge $q$ (right) for the test dataset. The confidence bands refer to 1 standard deviation from the mean.

basin which accumulates water in a downstream area (in the bottom left of the domain) without further propagation, which are not sufficiently represented in the training domain.

We then performed a fine-tuning step consisting of training again the previously-trained model with one extra simulation from the new case study. We trained and validated on the same simulation since we wanted to minimize the amount of data needed to fine tune the model. While in principle this might lead to overfitting, it was not the case here. This is probably due to the inductive biases of the model which constrain the model to learning only local dynamics. Table 2 shows that adding just one simulation improves the testing performance on the rest of the dike ring by $158\%$ and $62\%$ in MAE for water depth and discharge, respectively, and $38\%$ and $78\%$ for $\mathrm{CSI}_{0.05m}$ and $\mathrm{CSI}_{0.3m}$.

Figure 9 shows the model performance for the prediction in time of water depth for one test case. Water depths are overall well predicted in the domain, including water accumulation in the western part of the area. Errors are close to the breach location or flood front, mostly towards the end of the simulation. While the absolute values of the difference may be relatively

**Table 2.** Effect of fine-tuning the mSWE-GNN model on dike ring 15. The provided uncertainty estimates account for the variability across different simulations.

| Fine-tuning | MAE ↓ | | $\mathrm{CSI}_\tau$ [%] ↑ | |
|---|---|---|---|---|
| | h $(m)$ $[10^{-2}]$ | $|q|$ $(m^2/s)$ $[10^{-2}]$ | $\tau$=0.05 $m$ | $\tau$=0.3 $m$ |
| No | $31.09 \pm 5.42$ | $3.37 \pm 1.24$ | $63.36 \pm 19.54$ | $46.06 \pm 18.62$ |
| Yes | $12.07 \pm 4.19$ | $2.08 \pm 0.82$ | $87.68 \pm 10.3$ | $81.82 \pm 16.07$ |



**Figure 9.** mSWE-GNN's predictions for water depth on a testing simulation for dike ring 15. The topography is presented in the top left plot, the discharge hydrograph in the top right, and below the evolution over time for ground-truth output of the numerical simulation (top row) with the predictions (middle row). The difference (bottom row) is evaluated as the predicted value minus the ground-truth one; thus, positive values correspond to model over-predictions while negative values correspond to under-predictions.





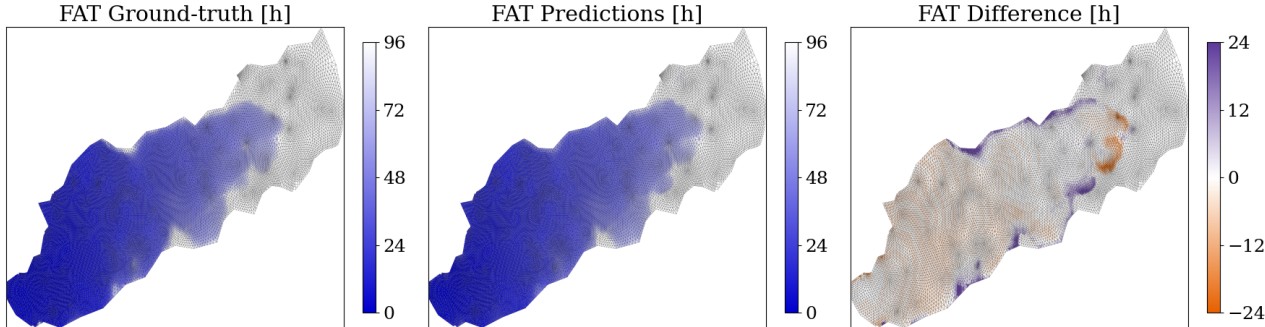

**Figure 10.** Flood arrival times (FAT) for a water depth threshold of $0.05m$ for a test case from dike ring 15, given for the numerical simulation (left), the predictions (center), and the difference (right). Darker colors in the first two maps indicate a faster arrival of the water, while white cells indicate absence of water. In the difference map, positive values correspond to model over-predictions while negative values correspond to under-predictions.

high in these areas, they do not matter as much for practical purposes since those locations are either way flooded with a high water depth, thus the associated damages will be equivalent.

Figure 10 shows that the spatio-temporal evolution of the predicted flood is in line with the corresponding numerical simulations, as indicated by the low errors of flood arrival times (FAT) for the critical threshold of 0.05 $m$ of water depth. FAT indicate the arrival time of water with a given depth threshold, for each cell in the domain. Most errors are located at the wave front during the end of the simulation, as previously mentioned, or in false positive areas that are not flooded in the numerical model.

Figure 11 indicates that the model performance is consistently high for all testing breach locations of the dike ring 15 dateset, as suggested by the high $\text{CSI}_{0.05m}$ values, which are always above 0.8. One reason why the model performs so well is that the

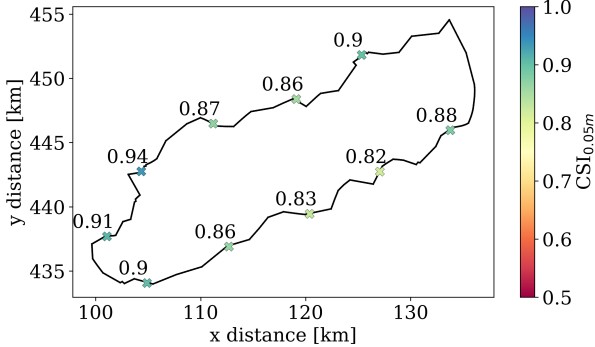

**Figure 11.** Performance in terms of CSI for a water depth of $0.05m$ for all testing breach locations in dike ring 15, for the fine-tuned mSWE-GNN.



**Table 3.** Ablation study on the removal or addition of individual architectural and training components, for the synthetic testing dataset. These are: using a learnable pooling for the down-sampling operator, removing skip connections in Eq. (7), removing the 1D-CNN in Eq. (8), and using rotation-dependent inputs. The best results are reported in **bold**.

| | DL model | MAE ↓ | | CSI$_\tau$ [%] ↑ | |
|---|---|---|---|---|---|
| | | h $(m)$ $[10^{-2}]$ | $|q|$ $(m^2/s)$ $[10^{-2}]$ | $\tau$=0.05 $m$ | $\tau$=0.3 $m$ |
| | SWE-GNN | $9.52 \pm 5.03$ | $0.42 \pm 0.16$ | $68.7 \pm 18.9$ | $51.7 \pm 22.1$ |
| | mSWE-GNN | $\mathbf{4.84 \pm 2.3}$ | $\mathbf{0.27 \pm 0.13}$ | $\mathbf{84.02 \pm 9.18}$ | $\mathbf{69.56 \pm 17.25}$ |
| mSWE-GNN | with learnable pooling | $5.72 \pm 3.09$ | $0.32 \pm 0.13$ | $81.23 \pm 12.23$ | $63.67 \pm 19.66$ |
| | w/o skip connections | $5.22 \pm 2.22$ | $0.32 \pm 0.15$ | $82.44 \pm 10.82$ | $66.81 \pm 17.31$ |
| | w/o 1D-CNN | $5.57 \pm 2.5$ | $0.32 \pm 0.14$ | $80.75 \pm 10.83$ | $65.03 \pm 19.21$ |
| | w/o rotation invariant inputs | $6.07 \pm 2.27$ | $0.34 \pm 0.15$ | $79.93 \pm 10.18$ | $62.89 \pm 18.28$ |

final flood map tends to converge to the downslope accumulation area in the bottom left area of the domain. This also proves that the model can correctly model the response dynamics of the system, independently of where the breaching starts.

The good performance of the mSWE-GNN model is accompanied by a substantial speed-up of the underlying model. When testing, the model has a speedup of more than 700 times with respect to the original simulations, as highlighted in Table A1. This indicates a good scaling with the size of the domain, with higher speed-ups for bigger domains. One other possible explanation is related to the simulated discharges. Numerical simulations of slow flows are generally more stable and faster to compute than those with high Froude numbers, which are more present in dike ring 15. Contrarily to numerical models, the mSWE-GNN has no such stability constraints, which make it unbound by the same limitations and thus faster.

## 4.3 Ablation study

Finally, we performed an ablation study to determine the role of the different components in the mSWE-GNN (Table 3), such as the multi-scale module, the convolutional decoder, and the rotation invariant inputs. We also reported the performance of the best SWE-GNN model from Section 4.1. The results reported in Table 3 show that all of the added or removed components contribute to the performance on the test dataset. The speed-up was consistent throughout all mSWE-GNN configurations and we report it in Table A1.

**Multi-scale module.** We analysed the effects of using a learnable down-sampling operator in place of a mean pooling in Eq. (5) and removing skip connections in Eq. (7). For the learnable down-sampling operator, we used a 3-layer MLP shared across each intra-scale edge that takes as inputs the dynamic node feature at nodes $i$ in $\mathcal{M}_m$ and node $j$ in $\mathcal{M}_{m+1}$, similarly to Eq. (6).





Using a learned down-sampling operator results in a lower performance. We argue this is caused by the unnecessary com­plexity of the operation and due to the common mean aggregation term, which is needed to make the model work with a flexible number of nodes, that cancels the expressivity of the MLP.

Removing skip connections does not influence as much the performance. This indicates that most of the computations are performed after the architecture bottleneck, while the down-going branch is responsible for smaller details that are not captured

in the up-going branch. This means that the number of layers in the down-going branch can probably be reduced, while keeping good performance with less model complexity.

**Decoder.** We compared the convolutional decoder (Eq. (8)) with a residual connection which simply sums the output of the previous time step to the output of the decoder's MLP before applying a $ReLU$ activation, i.e., $\hat{\mathbf{U}}^{t+1} = \sigma\left(\mathbf{U}^t + \varphi\left(\mathbf{H}_d^{(L)}\right)\right)$. Using the 1D-CNN in the decoder results in better testing and validation metrics, meaning that different time steps contribute

unevenly to the final model output. This allows the model to better capture variations in time, especially due to rapid variations in boundary conditions.

**Rotation invariant inputs.** We added the $x$ and $y$ components of the slope and orientation of mesh edges as static inputs to show that including rotation-dependent inputs worsens generalization (Table 3). The reason for this is that all simulations are quite different one from the other in terms of breach location and orientation of the meshes. Consequently, a model with

rotation-dependant inputs would require much more training data to generalize well to all spatial configurations.

## 5   Discussion

We proposed a multi-scale graph neural network model (mSWE-GNN) that can generalize flood simulations to unseen irregular meshes, topographies, and time-varying boundary conditions, with speed-ups up to 700 times compared to the underlying numerical model. The mSWE-GNN generalizes well to realistic case studies with as little as one fine-tuning simulation. This

result is in line with a similar finding for pluvial flooding where one fine-tuning simulation was enough to help generalization to diverse case studies (Cache et al., 2024). Since the model can generalize well with as little as 60 training simulations, we believe that training the model on a substantially larger amount of data might even remove the need of fine-tuning, although this could still be needed for more complex domains.

One key to the model's success are the different scales, which enable learning varying speeds of flood propagation and

capturing the hydraulic processes, contrary to the SWE-GNN, which learns a more limited range of speeds. The multi-scale nature of the model allows optimizing computations for areas where fine details are relevant only in small portions of the domain. In the same way, scales can be used to better include the presence of 1D structures in the domain such as channels and elevated elements. These can be included in the coarser meshes by using slimmer cells that overlap with the channel, as done in numerical models (Bomers et al., 2019). On the other hand, structures that markedly influence the flow propagation,

like levees, can simply be omitted in the coarser meshes, by leaving holes in correspondence of them. This artificially blocks the possibly faster flow propagation of coarser scales. Once over-topping of said levee occurs at the fine scale, then the faster propagation can begin anew in the coarser scales. We improved the model generalization to unseen meshes by considering





rotation-invariant inputs. This was possible because we considered scalar outputs, since we deemed the intensity of the flood more important than also knowing its direction for practical uses (Kreibich et al., 2009).

Regarding the process of mesh creation, we constructed the coarse-scale meshes based on the boundary polygon of the considered areas. However, this requires the user to create a mesh with a top-down approach and limits the use of an existing fine-scale mesh. This could be solved by using a different multi-scale mesh creation approach. For example, Lino et al. (2022) used a sampling strategy based on a regular partitioning of the domain, which allows the coarse meshes to have similar edge lengths, independently from the fine mesh. Alternatively, we could use the same mesh creation procedure to only generate the

coarse scale meshes and use existing detailed meshes in the fine scale. The latter may be problematic when fine structures are present that markedly alter the flow of the flood, making the automatic mesh generation procedure challenging.

    The boundary condition insertion technically works also for given water levels at the boundary, but we did not analyze it. Moreover, we did not analyse the performance for multiple concurring boundary conditions, despite the model can already accommodate them. To simplify the hyperparameter selection process, we also selected an equal number of GNN layers for

all scales. Instead, we could further optimize the Pareto front by changing the number of layers at each scale independently. Additionally, we did not compare with other recent developments in deep learning models, such as Fourier Neural Operators (Li et al., 2020) or Neural fields (Yin et al., 2023), since they either do not generalize across different irregular meshes or their application to flooding would not be trivial. We remark that most of the speed-ups come from the use of a GPU, as all processes are parallelizable. This is a well-known benefit of deep learning models and the mSWE-GNN enjoys it.

For practical applications, there are still several components that must be included to match numerical models for real case studies. Future studies should investigate the inclusion of time-varying breach growth models or components such as existing water bodies and linear elements, such as roads and secondary dikes. Eventually, the proposed model can be used to create a probabilistic framework to assess many different flood scenarios and uncertainties in boundary conditions, breaching conditions, and topography (Vorogushyn et al., 2010).

## 405   6   Conclusion

We proposed a multi-scale hydraulic graph neural network, called mSWE-GNN, that models flood propagation in space and time across multiple resolutions. The model takes as input static attributes, such as topography, and dynamic attributes, such as water depth and unit discharge at time $t$, and predicts their evolution at the following time step $t+1$. This is done via a U-shaped architecture that applies graph neural networks at different scales and combines them with down-sampling and up-sampling

operators. This captures a broader range of dynamics by jointly modelling the flood propagation speeds at different scales. We included time-varying boundary conditions via ghost cells. We also improved the generalization to unseen meshes by using rotation-independent inputs.

    The model can accurately replicate the overall dynamics of the flood evolution over unseen meshes, topographies, and boundary conditions, with no dependence from any numerical solver. The model can also generalize to realistic case studies with

more complex and bigger domains than the training ones with only a single fine-tuning simulation. Moreover, the new model





is better and faster than its non-multi-scale counterpart, indicating that the insertion of this module contributes significantly to the model's performance.

Overall, these results open up new possibilities to model probabilistically flood uncertainties in real case studies. This will allow practitioners to have a complementary tool for fast evaluation of several flooding scenarios before analysing more in
depth critical ones with numerical models.

## Appendix A:  Supplementary results

We analysed the evolution in space and time of the unit discharges for one test simulation in the synthetic dataset, to highlight that the model is now able to correctly model the filling and emptying dynamics. Figure A1 shows that discharges are modelled very well by the model, both in the ascending and the descending phases of the input hydrograph. This is in line with the hy-
pothesis of Bentivoglio et al. (2023) according to which the model is able to capture these draining and decreases in discharges when presented sufficient samples of it in the training dataset.

We report the execution run times of the numerical and trained mSWE-GNN models for both testing datasets. Since the deep learning model is run in parallel, the prediction times per simulation are averaged out through all simulations. We measure the run time variability by running the model 10 times and reporting the corresponding mean and standard deviation. For both
dataset, the model achieves a great speed-up, of more than two orders of magnitude, which could further increase when selecting a smaller model from the Pareto front in Figure 7.

In terms of training times, the SWE-GNN model took between 5 to 30 hours while the mSWE-GNN 2 to 15 hours, depending on the model complexity. The fine-tuning process, with the selected mSWE-GNN model in Section 4.1, took around 20 minutes. The fine-tuning time can be reduced to 5 minutes by decreasing the number of epochs, while still obtaining comparable
performance. If we evaluate the speed-up on dike ring 15 including also the time to run the fine-tuning numerical simulation and the time to train it, we still achieve a speed-up of 4 to 8 times, depending on the number of fine-tuning epochs.

## Appendix B:  Mass conservation

We proposed a regularization term $\mathcal{L}_c$ that enforces a global mass conservation per each time step. This reads as

$$\mathcal{L}_c = \left| \sum_{i=1}^{N} a_i \Delta \hat{h}_i - Q \Delta t \right| \tag{B1}$$

| Dataset | Numerical model [$s$] | mSWE-GNN [$s$] | Speed-up [-] |
|---------|----------------------|----------------|--------------|
| Test | $80.8 \pm 15.4$ | $0.33 \pm 0.10$ | $250 \pm 25$ |
| Test dike ring 15 | $611 \pm 211$ | $0.81 \pm 0.23$ | $750 \pm 50$ |

**Table A1.** Run times of the numerical model and the selected mSWE-GNN model for the two testing datasets and their respective speed-ups.





**Figure A1.** mSWE-GNN's predictions for unit discharges a test simulation from the synthetic dataset. The evolution over time for ground-truth output of the numerical simulation (top row) with the predictions (middle row) are represented using a logarithmic scale to better appreciate the values' distribution. The difference (bottom row) is evaluated as the predicted value minus the ground-truth one and is kept with a standard scale to highlight the use of the logarithmic scale; positive values correspond to model over-predictions while negative values correspond to under-predictions. The legends refer to the maximum values throughout the whole simulation.





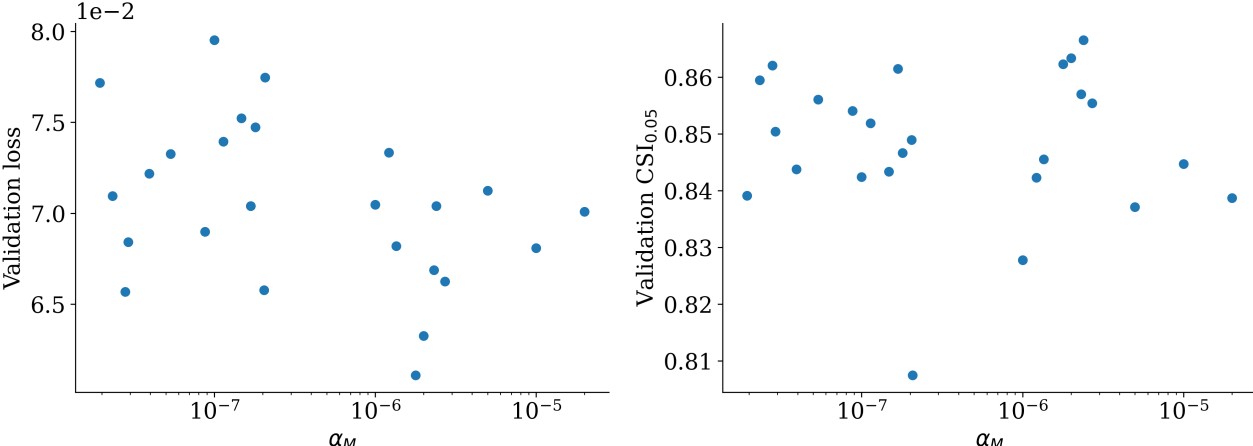

**Figure B1.** Performance in terms of validation loss and $CSI_{0.05m}$, for varying values of the mass conservation weight $\alpha_M$.

where $N$ is the number of nodes in the output mesh, $\Delta\hat{h}_i$ is the variation in predicted water depth at node $i$, $a_i\Delta\hat{h}_i$ is the variation in predicted volume at node $i$, $Q$ is the inflow discharge, and $\Delta t$ is the time interval between $t$ and $t+1$, in which the discharge is assumed to be constant. This enforces the total amount of volume entering the domain $Q\Delta t$ to be redistributed in the domain so that the volume is conserved.

We carried out supplementary experiments to explore the benefit of adding this term to the forecasting loss in Eq. (9). The
combined loss $\mathcal{L}$ can be expressed as

$$\mathcal{L} = \mathcal{L}_f + \alpha * \mathcal{L}_c, \tag{B2}$$

where $\alpha$ is weighs the contribution of the mass conservation term.

Figure B1 shows that the validation loss and CSI are slightly negatively correlated with $\alpha$, meaning that losses tend to improve and classification worsen. The reason why losses slightly improve might be because the added loss term depends only
on the predicted water depth, so it enforces that value to be more precise. However, the conservation loss acts globally for each time step, instead of locally. So, the model cannot correctly improve the spreading of the flood but only the absolute values of total water depth.

From these plots, we cannot extract any meaningful conclusion since there is no statistical significance, as highlighted by p-values of 0.42 and 0.48, respectively. Moreover, the performance in the testing dataset follows an opposite trend, further
indicating that inclusion of this term is not consistently better. This in part contradicts the idea of physics-informed neural networks (PINN), according to which adding a physical loss term improves performance (Raissi et al., 2019). One motivation is that the loss we employ does not rely on auto-differentiation in the same way that PINN do. We also evaluate it globally, rather than at individual points as in PINNs. Nonetheless, this loss is independent of the ground-truth data, making it a possible self-supervised loss that could be explored in future works.



## Appendix C: Parallel simulations

We refactored the code to compute all testing simulations in parallel, instead of in series, by using batches. To analyse the speed-ups provided by parallel execution, we selected all models in the Pareto front of the mSWE-GNN in Figure 7, which have different number of parameters and number of GNN layers per scale. We then ran the models using an increasing number of simulations in parallel, indicated by the batch size.

Figure C1 indicates that the speed-up almost doubles with the batch size, independently of the size of the model. It also highlights that the main computational effort comes from the an increase number of GNN layers, rather than just the total number of model parameters, as also reported in Bentivoglio et al. (2023). When running 20 simulations, i.e., the full testing dataset, in parallel instead of in series, we provide a further speed-up of 4.5 times on average across different models' sizes. Further speed-ups may be achievable by optimizing the code; for example just-in-time(JIT)-compiling of the PyTorch code into optimized kernels can further accelerate the execution of the model by two or three times (Paszke et al., 2019). In a similar fashion, IPUs, which are a novel processing unit that has faster inference on graphs, can further speed-up the model by two to four times (Knowles, 2021).

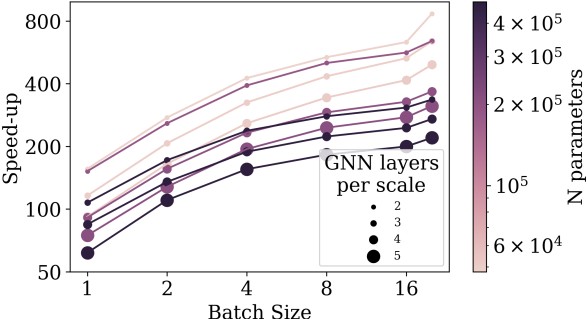

**Figure C1.** Speed-ups of mSWE-GNN for the synthetic test dataset, considering varying batch sizes, i.e., how many simulations are run in parallel. The results are reported for all Pareto front models from Figure 7. Both axes are in log(2) scale.

## Appendix D: Hyperparameter ranges

We reported the hyperparameters used to create and train the model and their ranges in Table D1. Since the amount of hyper-parameters is high, some values are taken based on similar studies in literature (e.g., Bentivoglio et al., 2023). For the mass conservation weight $\alpha$ in Eq. (B1), we uniformly sampled values in an interval from $10^{-8}$ to $5 * 10^{-5}$, using a logarithmic distribution. The reason why these values are small is due to the flood volumes being more than $10^6$ times higher than water depths.





**Table D1.** Summary of the hyperparameters and related values' ranges employed for the different deep learning models. The **bold** values indicate the best configuration in terms of validation loss.

| DL model | Hyperparameter name | Values' range (**best**) |
|---|---|---|
| All models | Initial learning rate | 0.003 |
| | Input previous time steps ($p$) | 2 |
| | Maximum training steps ahead ($H$) | 6 |
| | Optimizer | Adam |
| | Batch size | 12 |
| | $\alpha$ | **0**, $[10^{-8}$-$5*10^{-5}]$ |
| SWE-GNN | Embedding dimension ($G$) | 16,32,50,**64** |
| | Number of GNN layers ($L$) | 10, 12, 14, **16**, 18 |
| mSWE-GNN | Embedding dimension ($G$) | 16,32,50,**64** |
| | Number of GNN layers ($L$) | 2,3,**4**,5 |

*Code and data availability.* The employed dataset can be found at https://dx.doi.org/10.5281/zenodo.13326595. The code repository will be
480  available after publication.

*Author contributions.* **Roberto Bentivoglio**: Conceptualization, Methodology, Software, Validation, Data curation, Writing- Original draft preparation, Visualization, Writing - Review & Editing. **Elvin Isufi**: Supervision, Methodology, Writing - Review & Editing, Funding acquisition. **Sebastiaan Nicolas Jonkman**: Supervision, Writing - Review & Editing. **Riccardo Taormina**: Conceptualization, Supervision, Writing - Review & Editing, Funding acquisition, Project administration.

485  *Competing interests.* No competing interests are present.

*Acknowledgements.* This work is supported by the TU Delft AI Initiative program. We thank Deltares for providing the license for Delft3D to run the numerical simulations.



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
