# Peer review of "Multi-scale hydraulic graph neural networks for flood modelling"

_EGUsphere, 2024_

## Author Comment (AC1)

We thank the Reviewer for the helpful comments and suggestions. We hereby address them individually. In this document we indicate the Reviewer's comments in *italic dark grey*, while text that was changed in the paper in blue.

*This paper presents a deep-learning based approach for 2D surface flood modelling that builds on a previously existing model, SWE-GNN. It proposes an alternative model that overcomes the need of a numerical solver to determine initial conditions and includes other novelties to improve the speed and generalization of the model. The paper demonstrates that the model can benefit from fine-tuning to generalize to new case studies. In my opinion, the research is interesting and is very well presented. I am therefore recommending the paper for publication, after minor revisions.*

**General comments**

*The paper proposes a model for flood modelling and is applied only to a specific case of floods; dike-breach floods. It would be nice to have a section in the discussion on how the model could apply to other kind of floods, such as fluvial and pluvial floods for example.*

We thank the Reviewer for the valuable point addressed here. We added a paragraph in the discussion section that indicates how to deal with other types of floods. This is written in lines 390-394 as:

"While the current model framework can work for dike-breach floods, we did not evaluate it for other types of floods. For river and costal floods, the model should work as well without any changes since the inputs are of the same type as dike breach floods, e.g., upstream discharge hydrograph or sea water levels. On the other hand, pluvial floods require precipitation as a further input. Assuming rainfall as a spatially distributed variable, it could be added as a dynamic forcing; this could work in a similar way as for static features, but changing at each time step, independently of the predicted output. For urban floods, the drainage system should also be included. This could be done, as in numerical methods, by coupling the overland flow, predicted by the mSWE-GNN, with a 1D model for the sewers, possibly with another learned GNN as in Garzon et al., 2024."

**Specific comment**

*L84 – L85: For reproducibility, I would suggest clarifying how a mesh is classified as being too small and how the resolution of the fine mesh is chosen.*

We thank the Reviewer for pointing out this concern. While addressing it, we also noticed a writing mistake, since "mesh edge" should be "flow edge", which are defined in the MeshKernel library as "the edges connecting faces circumcenters". Hence we corrected the corresponding sentence and also expanded on the definition of small mesh and mesh resolution.

Lines 83-86 now read as:

"For the same numerical constraints, after the orthogonalization, all elongated elements get removed, resulting in a mixture of triangular and quadrilateral elements. We define elongated elements as those whose line connecting barycentre and edge middle points is 0.1 times smaller than the other lines in the same element."

*L93: It might be better to define what epsilon is here rather than in L104.*

We thank the Reviewer for spotting this mistake. The epsilon in line 93 is not the same as the one in line 104. For this reason, we removed the first epsilon and rephrased the corresponding sentence as: "...if edge (i, j) exists."

*Eq. 3 and 4: I think that it might be useful to have the clarification of what h_di and h_si are.*

We added in line 114 a clarification of what these embedding mean:

"The encoded variables Hs, Hd, and E' represent a higher-dimensional version of the original inputs that is more expressive."

We also clarified what h_di and h_si are in lines 135-136 as:

"where $\psi(\cdot) : R^{5G} \rightarrow R^G$ is an MLP, $\odot$ is the Hadamard (element-wise) product, $h^{(\ell)}_{di}$ is the embedding of the dynamic inputs at node I and layer $\ell$, h_si is the embedding of the static inputs at node i, and $W^{(\ell)} \in R^{G \times G}$ are learnable parameter matrices."

*L190: I would suggest replacing 'training simulations' by 'training data' here for clarity.*

We replaced the term as suggested.

*L193: You should explain that water level is the sum of the water depth and elevation of the cell as this is might not be straightforward.*

We clarified how water levels are calculated in lines 193-194 as:

"... and w_i^t its water level, given by the sum of the elevation and water depth at time t."

We clarified the meaning of the variable O in lines 206-207 as:

"... H is the prediction horizon, O the number of output hydraulic variables, and gamma_o are coefficients used to weigh the influence of each hydraulic variable to the loss."

*Table 1: To which resolution of the mesh does the edge length refer?*

We included an additional explanation line in the caption of Table 1 that reads as:

"All geometric variables refer to the properties of the finest mesh in each dataset."

*Section 3.1: I believe a concise explanation of how the Manning's n coefficients are defined for the synthetic dataset would be helpful. I'm curious to know if these coefficients are spatially variable as the model might face challenges in extracting meaningful insights from them if they were spatially uniform.*

We thank the Reviewer for the interesting point. In our experiments, we kept the same, spatially-uniform, roughness coefficients for all simulations, using a value of 0.023 $s/m^{1/3}$ everywhere, despite the model can potentially work with different values of it. We included this clarification in lines 219-221 as:

"For the Manning's roughness coefficient m, we used a spatially uniform value of 0.023 $m^{1/3}$ $s^{-1}$, which is kept the same throughout all simulations."

In the discussion section, we also added a further paragraph discussing possible implications of using a spatially varied distribution of roughness values (lines 402-403):

"We employed a constant and spatially uniform roughness coefficient, meaning that we did not assess how the model generalizes to different values and spatial distributions. This might lead to different dynamics that, following the same reasoning as for the different speeds of propagation, the model should still be able to capture."

*L270-275: You provide a nice and clear explanation of CSI. However, you consider 2 thresholds (0.05 m and 0.3 m) while you explain in L273-275 that the flooded and non-flooded cells are distinguished from another in TP, FP and FN. However, this isn't*

*entirely accurate as cells with water depths below 0.3 m could still be considered as flooded. I would try to be more accurate and write for example: 'TP are true positives, i.e. number of cells where both model and simulations predict water levels above the threshold value'. The same applies for the description of FP and FN.*

We adapted the suggestion to lines 276-278 as:

"where TP are the true positives, i.e., the number of cells where both numerical and deep learning models predict water depth above a given threshold, FP are the false positives, i.e., the number of cells where the deep learning model wrongly predicts water depth above a given threshold, and FN are the false negatives, i.e., the number of cells where the deep learning model does not predict water depth above a given threshold."

*L285: I would suggest adding a sentence stating that you are assuming the enhanced SWE-GNN also outperforms the other models, given that the original SWE-GNN outperforms them and despite the added modifications.*

We thank the Reviewer for the comment. We added a sentence to clarify that the enhanced SWE-GNN model should also outperform other models, in lines 288-289:

"We also did not compare against other baselines as the SWE-GNN performs better than them (Bentivoglio et al., 2023), so we assumed the same holds for the enhanced version."

*L305: Do you know why the MAE of h increases? Could it be due to error propagation throughout the simulation? If you have any insights into the cause of this increase, it might be worth adding it.*

We thank the Reviewer for the valuable suggestion. We added the main insight on why errors tend to increase in time, in lines 308-309:

"The main reason for the increase in water depth MAE over time is that as the flood progresses, it covers a greater spatial extent, increasing the number of cells where prediction errors can occur."

*Fig. 9 and Fig. 10: Consider using different color scales as it is difficult to visualize the differences in water levels as it is. Also, could you clarify which mesh (e.g. finest mesh) is shown in the figures?*

We clarified in figures 9 and 10 that the plots refer to the finest mesh by adding the following sentence in the figure captions:

"All plots represent values only on the finest mesh."

Regarding the color scales, we believe that the employed scales provide a good combination of colors to highlight the results of water depths, flood arrival times, and their differences. We recognize that some colors might not be visible because of the overlap with the mesh so we improved the figures by reducing the width of the computational mesh edges, which allows to better see the colors.

We report the change in Figures 9 and 10 below.

[Figure]

Figure 10

[Figure]

Figure 9

*Fig. 10: It is really nice that you detail what the negative and positive values correspond to in the captions of Fig. 9 and Fig. 10. However, I would recommend revising the caption in Fig. 10 to enhance clarity, e.g. 'positive values indicate that the model estimates later arrival times than the numerical simulation, while negative values indicate that the model predicts earlier arrival times'*

We thank the Reviewer for the suggestion. We changed the caption as recommended.

**Technical corrections**

We thank the Reviewer for spotting these issues. We agree with all of them and have modified them as suggested. In case of modifications that are not exactly as suggested, we clarified why and how after the comment.

*Throughout the manuscript (e.g., L14, L51, L223, caption of Fig.5): Uncapitalize 'the' in 'The Netherlands'*

*L178: 'via edges directed towards' rather than 'via directed edges towards'*

*L292: Clarify 'is comparatively faster than the numerical model'*

We clarified the sentence by removing the term "comparatively" which indeed was adding unnecessary confusion.

*Throughout the manuscript: Add a spacing between the numbers and the 'm' for meters*

*L302-303 and 305: Replace the four 'in correspondence of' with 'occur at', 'occur simultaneously to', 'near' and 'at' respectively*

We replaced all four 'in correspondence of' as suggested, except the second one which was changed to "close to".

*Fig. 8: You might want to keep the notation of CSI0.05 and CSI0.3 like in the manuscript, i.e. add the 'm'*

*Table 2 and 3: Write the units as [10-2 m] and [10-2 m2/s]*

*L330: Typo in 'dataset'*

*L365: Typo in 'dependent'*

*L393: 'analyze' rather than 'analyse' to keep consistency with the previous sentence*

*L466: Remove 'the' in 'comes from the an increase'*

References:

Garzón, A., Kapelan, Z., Langeveld, J. and Taormina, R., 2024. Transferable and data efficient metamodeling of storm water system nodal depths using auto-regressive graph neural networks. *Water Research*, p.122396.

---

## Author Comment (AC2)

We thank the Reviewer for the helpful comments and suggestions. We hereby address them individually. In this document we indicate the Reviewer's comments in *italic dark grey*, while text that was changed in the paper in blue.

*This paper presents a multi-scale, hydraulic-based graph neural network model (mSWE-GNN) designed to enhance flood modelling. Building on previous models like SWE-GNN, the authors introduce novel elements such as rotation-invariant inputs, ghost cells for boundary conditions, and a multi-scale architecture to simulate flood events over variable topographies and unseen meshes. The paper includes extensive theoretical explanations and benchmarking evidence that demonstrate the model's effectiveness across different flood scenarios. Overall, this research is timely and relevant, contributing meaningfully to the field of flood modeling with machine learning. I support the publication of this paper following some revisions.*

**General comments**

*The model presented in this paper, mSWE-GNN, demonstrates impressive capabilities in simulating dike-breach flood scenarios specifically. However, its applicability appears limited to this particular type of flood. Given the diversity of flood types—such as fluvial and pluvial floods—extending the model's capabilities or discussing potential adaptations for these other scenarios would significantly enhance its relevance and utility. A broader application across flood types would open up further opportunities for practical deployment and highlight the robustness of the proposed multi-scale architecture in different hydraulic contexts. Including a discussion on how the model might be adapted for other flood types would be a valuable addition.*

*However, the paper limits its application to dike-breach floods and does not explore other flood scenarios. Adding a section discussing potential adaptations for other flood types, such as fluvial or pluvial floods, would strengthen the paper's relevance.*

We thank the Reviewer for the valuable point addressed here. We added a paragraph in the discussion section that indicates how to deal with other types of floods. This is written in lines 390-394 as:

"While the current model framework can work for dike-breach floods, we did not evaluate it for other types of floods. For river and costal floods, the model should also perform well without modifications since the inputs are of the same type as dike breach floods, e.g., upstream discharge hydrograph or sea water levels. On the other hand, pluvial floods require precipitation as a further input. Assuming rainfall as a spatially distributed variable, it could be added as a dynamic forcing; this could work in a similar way as for static features, but changing at each time step, independently of the predicted output. For urban floods, the drainage system should also be included.

Similarly to numerical methods, this could be achieved by coupling the overland flow predicted by the mSWE-GNN with a 1D model for the sewers, potentially using another learned GNN as demonstrated in Garzon et al., 2024."

References:

Garzón, A., Kapelan, Z., Langeveld, J. and Taormina, R., 2024. Transferable and data efficient metamodeling of storm water system nodal depths using auto-regressive graph neural networks. *Water Research*, p.122396.

*Additionally, the inclusion of a mass conservation term in the loss function, though discussed in the appendix, warrants further investigation, particularly regarding its influence on model stability.*

*Discussion on Physical Constraints (Appendix B): The addition of mass conservation loss is an interesting consideration. However, it would be helpful if the authors could provide further analysis or reasoning on why this term, although theoretically sound, did not yield a consistent improvement in testing. This section could benefit from a comparison of other physical constraint approaches in neural networks.*

We thank the Reviewer for the feedback. We gave an explanation of why we believe that the additional loss term did not improve our performance in lines 449-452:

"The reason why losses slightly improve might be because the added loss term depends only on the predicted water depth, so it enforces that value to be more precise. However, the conservation loss acts globally for each time step, instead of locally. So, the model cannot correctly improve the spreading of the flood but only the absolute values of total water depth."

and in lines 456-458, in comparison with PINNs:

"One motivation is that the loss we employ does not rely on auto-differentiation in the same way that PINNs do. We also evaluate it globally, rather than at individual points as in PINNs."

We believe that there are no additional main points that could cause justify the absence of improvements, as we were also expecting the loss term to improve our results.

Regarding other physical constraint approaches in neural networks, there are three main ways to add them: 1) using PINNs, which rely on predicting the variables (and their derivatives, by means of auto-differentiation) present in the underlying partial differential equation (PDE) and minimize the corresponding loss in several collocation points; 2) using a regularization term in the loss function that works similarly to the

PINN's loss, but without using the original PDE; 3) embedding physics in the model's architecture.

The mSWE-GNN, thanks to the computational structure similar to numerical hydraulic models, enforces locality of flood propagation, thus providing the third type of physical constraint defined before. We avoided PINNs and decided to explore softer physical constraints since we would have had to change the model's inputs and outputs so that the auto-differentiation could provide us with the estimates of the derivatives with respect to the predicted target variables, needed for the shallow-water equations. Because of this, adding PINNs would go out of the scope of the paper, which was to show that adding multi-scale information improves the model performance, that we can include a wide range of boundary conditions via ghost cells, and that rotational-invariant inputs improve generalization. Moreover, PINNs are typically designed to solve a given physical problem for a single set of boundary and initial conditions. This approach does not inherently generalize across varying conditions, which is a prerogative of our work. We included this justification in lines 470-475 as:

"This in part contradicts the idea of physics-informed neural networks (PINN), according to which adding a physical loss term improves performance (Raissi et al., 2019). One motivation is that the loss we employ does not rely on auto-differentiation in the same way that PINNs do. We also evaluate it globally, rather than at individual points as in PINNs. Implementing a PINN loss would require adjustments to the model's inputs and outputs to allow auto-differentiation to estimate the derivatives of the predicted target variables. Moreover, PINNs are typically designed to solve a given physical problem for a single set of boundary and initial conditions, thus limiting the model's capacity to generalize across varying conditions, which is a prerogative of our work. Although we did not adopt this approach here, it could be explored in future studies. Notably, our loss term is independent of ground-truth data, making it a possible self-supervised loss that could be explored in future works."

**Specific comment**

*L89-92: For clarity and reproducibility, it would be beneficial for the authors to provide additional detail on the criteria for mesh size. Specifically, a brief explanation of how mesh resolution is chosen and classified as "too small" would enhance reproducibility.*

We thank the Reviewer for pointing out this concern. While addressing it, we also noticed a writing mistake, since "mesh edge" should be "flow edge", which are defined in the MeshKernel library as "the edges connecting faces circumcenters". Hence we corrected the corresponding sentence and also expanded on the definition of small mesh and mesh resolution.

Lines 83-86 now read as:

"For the same numerical constraints, after the orthogonalization, all elongated elements get removed, resulting in a mixture of triangular and quadrilateral elements. We define elongated elements as those whose line connecting barycentre and edge middle points is 0.1 times smaller than the other lines in the same element."

*Eq. 3 & Eq. 4: It may be helpful to specify definitions for hdih_{di}hdi and hsih_{si}hsi, as these variables are key in understanding the propagation rule and its application.*

We added in line 114 a clarification of what these embedding mean:

"The encoded variables Hs, Hd, and E' represent a higher-dimensional version of the original inputs that is more expressive."

We also clarified what h_di and h_si are in lines 135-136 as:

"where $\psi(\cdot) : R^{5G} \rightarrow R^G$ is an MLP, $\odot$ is the Hadamard (element-wise) product, $h^{(\ell)}_{di}$ is the embedding of the dynamic inputs at node I and layer $\ell$, $h_{si}$ is the embedding of the static inputs at node i, and $W^{(\ell)} \in R^{G \times G}$ are learnable parameter matrices."

*Boundary Condition Section (2.3): This section introduces ghost cells effectively but lacks explicit examples or visual representation of how they are implemented for various boundary types (e.g., inflows and outflows). Including this could aid in comprehending the technique's generalizability to different boundary configurations.*

We clarified figure 3 to better visualize how the type of boundary condition affects the direction of the edge between the ghost cell and the boundary cell. Figure 3 now shows as:

[Figure]

**Figure 3**

We also changed the caption to better describe this difference in type of boundary conditions as:

"Schematic representation of an arbitrary triangular volume mesh (left) with two ghost cells for inflow and outflow boundary conditions (BC). The ghost cells (red) are added in correspondence of a boundary cell which receives a given boundary condition. In the dual graph (right), a directed edge is added from the ghost cell to the domain cell, or vice-versa, depending on whether the boundary condition is an inflow or an outflow, respectively."

*Fine-Tuning Evaluation (Section 4.2): While the fine-tuning approach is well-motivated, it would be helpful to elaborate on the potential for overfitting given that a single simulation is used for fine-tuning and testing. The authors might explore this limitation and offer suggestions for future validation.*

We motivated why we selected only one simulation for training and validation in lines 315-317 as:

"We trained and validated on the same simulation since we wanted to minimize the amount of data needed to fine tune the model. While in principle this might lead to overfitting, it was not the case here. This is probably due to the inductive biases of the model which constrain the model to learning only local dynamics."

We added another motivation of why we argue this procedure does not lead to overfitting. In fact, the training process considers only a limited number of predicted steps ahead (H=6 in our experiments, as reported in line 255), while the full simulation has eight times more. This effect, combined with the previously-mentioned learning of local dynamics, makes it so that the model is forced to learn different dynamics in time rather than overfitting on a single pattern, even if we are using a single simulation.

A simple solution to the overfitting issue could also be to increase the size of the fine-tuning dataset, so that we further limit possible overfitting effects. In case of availability of more simulations at a low computational cost, this is surely a valuable alternative that should also improve the model's performance.

Accordingly, we further expanded lines 315-319 as:

"We trained and validated on the same simulation since we wanted to minimize the amount of data needed to fine tune the model. While in principle this might lead to overfitting, it was not the case here. This is probably due to the inductive biases of the model which constrain the model to learning only local dynamics. Additionally, the training process considers only a limited number of predicted steps ahead, while the full simulation has many more. Consequently, the model is forced to learn different

dynamics in time rather than overfitting on a single temporal pattern, even if we are training on a single simulation."

We also modified lines 374-376:

"The mSWE-GNN generalizes well to realistic case studies with as little as one fine-tuning simulation. We expect the model to further improve performance and reduce risk of overfitting by increasing the number of fine-tuning simulations. This result is in line with a similar finding for pluvial flooding where one fine-tuning simulation was enough to help generalization to diverse case studies"

*Table 2: To support interpretability, clarify which mesh resolution is referred to in this table. Also, in Table 3, presenting units consistently as [10−2 m][10^{-2} \text{ m}][10−2 m] and [10−2 m2/s][10^{-2} \text{ m}^2/\text{s}][10−2 m2/s] improves clarity and aligns with standard formatting practices.*

We clarified in Table 2 that all metrics refer only to the finest mesh by adding the following line to the caption "All metrics refer only to the finest mesh.".

We homogenized all units in the paper to have a consistent format.

*Appendix Figure A1: This figure is valuable in understanding unit discharge modeling, but it may benefit from a clearer representation of errors by using varying scales or alternative visual indicators for over- and under-predictions.*

The scale we employed in Figure A1 was automatically generated based on the minimum and the maximum errors found throughout the simulation. We changed these values and corrected accordingly the legend, to better show the errors distribution over time. We then removed "The legends refer to the maximum values throughout the whole simulation." From the caption. Figure A1 now shows as:

[Figure]

**Figure A1.** mSWE-GNN's predictions for unit discharges a test simulation from the synthetic dataset. The evolution over time for ground-truth output of the numerical simulation (top row) with the predictions (middle row) are represented using a logarithmic scale to better appreciate the values' distribution. The difference (bottom row) is evaluated as the predicted value minus the ground-truth one and is kept with a standard scale to highlight the use of the logarithmic scale; positive values correspond to model over-predictions while negative values correspond to under-predictions.

**Technical corrections**

*Throughout manuscript: Maintain consistent formatting of units, particularly with spacing between numbers and "m" for meters.*

We formatted all units in a consistent way, checking also for the spacing between numbers and units.

*L466: "comes from the an increase" should be corrected to remove the redundant article "the."*

We thank the Reviewer for spotting the typo, which we corrected accordingly.

*L302 & L305: The phrase "in correspondence of" could be replaced with more direct alternatives such as "occur at" or "align with" for clarity.*

We modified the two instances of "in correspondence of" respectively with "occur at" and "at the end".

**Conclusion**

*In summary, the paper presents an interesting deep learning model with clear advancements over existing approaches in flood modeling. The mSWE-GNN's ability to generalize and efficiently process multi-scale data is convincingly demonstrated. Nevertheless, a limitation of the current model is its focus on simulating dike-breach floods alone. Given the range of flood types—such as fluvial and pluvial floods—future work extending this model's application to other types would significantly increase its value and utility. Including a discussion on potential adaptations for different flood scenarios would further enrich the paper, offering insights into the generalizability and versatility of the multi-scale approach.*